# Urinary Porphyrin Profiles and Trace Element Imbalances in Children with Autism Spectrum Disorders: Insights into Environmental and Metabolic Biomarkers

**DOI:** 10.3390/ijms262110478

**Published:** 2025-10-28

**Authors:** Joško Osredkar, Kristina Kumer, Maja Jekovec Vrhovšek, Lidija Čuturić, Alenka France Štiglic, Teja Fabjan

**Affiliations:** 1Institute of Clinical Chemistry and Biochemistry, University Medical Centre Ljubljana, Zaloška Cesta 2, 1000 Ljubljana, Slovenia; josko.osredkar@kclj.si (J.O.); lidija.cuturic@gmail.com (L.Č.); alenka.stiglic@kclj.si (A.F.Š.); 2Faculty of Pharmacy, University of Ljubljana, Aškerčeva 7, 1000 Ljubljana, Slovenia; 3Center for Autism, Unit of Child Psychiatry, University Children’s Hospital, University Medical Centre Ljubljana, Mokrška Ulica 16C, 1000 Ljubljana, Slovenia; maja.jekovec@kclj.si

**Keywords:** autism spectrum disorder, porphyrins, lead, heme biosynthesis, environmental toxicants, biomarkers, trace elements, ICP-MS

## Abstract

Porphyrins are intermediates in heme biosynthesis and have been proposed as biomarkers of metabolic dysfunction and environmental exposure in autism spectrum disorder (ASD). This study aimed to evaluate urinary porphyrin fractions and trace element ratios in children with ASD compared to neurotypical controls. Urinary porphyrins were quantified using high-performance liquid chromatography (HPLC), and trace elements were measured via inductively coupled plasma mass spectrometry (ICP-MS) normalized to urinary creatinine. Trace element ratios (e.g., Zn/Cu, Se/Pb) were calculated. Statistical comparisons were made using the Mann–Whitney U-test. Children with ASD showed significantly elevated urinary levels of coproporphyrin (median: 1.94 µg/g creatinine vs. 1.32 in controls; *p* = 0.02) and pentacarboxyporphyrin (0.86 vs. 0.57; *p* = 0.01), and reduced hexacarboxyporphyrin (0.12 vs. 0.23; *p* = 0.03). Lead (Pb) levels were significantly higher in ASD (median: 1.96 µg/g creatinine vs. 0.82; *p* = 0.004), while mercury (Hg) was not significantly different. Several trace element ratios differed significantly: Zn/Cu (ASD 41.9 vs. controls 49.1; *p* = 0.021), Se/Pb (12.9 vs. 25.7; *p* = 0.002), Cu/Se (0.49 vs. 0.38; *p* = 0.008), and Zn/Pb (19.5 vs. 44.8; *p* = 0.002). The Hg/Se ratio did not differ significantly.: Children with ASD demonstrate altered porphyrin profiles and trace element imbalances, including increased Pb and disrupted Zn/Cu and Se/Pb ratios, indicating oxidative stress and impaired detoxification. Combined assessment of porphyrins and trace element ratios may provide valuable non-invasive biomarkers for environmental and metabolic disturbances in ASD.

## 1. Introduction

ASD is a complex neurodevelopmental disorder that is characterized by repetitive behaviors, limited interests, and difficulties with social communication. Clinical presentations are very diverse, with symptoms ranging from moderate to severe. Coexisting disorders like immunological dysregulation, behavioral issues, and gastrointestinal difficulties are frequently present as well [1]. ASD has a consistent diagnostic framework provided by the Diagnostic and Statistical Manual of Mental Disorders (DSM-5). The DSM-5 outlines core diagnostic criteria but does not elaborate on etiology. The multifactorial nature of ASD is increasingly attributed to genetic and environmental interactions [2,3].

Although several genes linked to ASD have been discovered in recent years, no single genetic mutation has been found to explain the disorder’s diverse clinical presentations. Furthermore, the significance of genetic contributions has been highlighted by the improvement in diagnostic accuracy brought about by comparative genomic array hybridization and other cutting-edge genomic approaches [3]. However, it is becoming more widely acknowledged that environmental exposures during crucial developmental windows, such as pregnancy and early infancy, play a significant role in raising the risk of ASD. The pathophysiology of ASD is thought to be significantly influenced by oxidative stress, DNA damage, and altered protein activity that might result from these exposures [1,4].

Among environmental toxicants, heavy metals such as lead (Pb) and mercury (Hg) are particularly relevant due to their known neurotoxicity and ability to disrupt cellular metabolism [5,6,7]. Both Pb and Hg interfere with enzymes involved in heme biosynthesis, leading to the accumulation of porphyrin intermediates in biological fluids [8].

Cyclic tetrapyrroles called porphyrins, which are essential for heme production, have been identified as possible indicators of metabolic abnormalities and environmental toxicity in ASD [9]. Heme is an essential molecule for oxygen transport, neurotransmitter control, and detoxification. Heme is produced by a sequence of enzymatic processes that result in porphyrin intermediates. Porphyrin precursors undergo a strictly controlled stepwise conversion as part of the heme biosynthesis pathway. Protoporphyrin and heme are the end products of a series of enzymatic decarboxylations that begin with uroporphyrinogen III and proceed to heptacarboxy-, hexacarboxy-, pentacarboxy-, and coproporphyrinogen III. This stepwise enzymatic pathway is well-characterized in porphyrin metabolism studies [9,10,11]. When the route is disturbed, which frequently happens as a result of metabolic stress or exposure to toxins (such as heavy metals like mercury), each of these intermediates can build up and be eliminated in urine. In the context of environmental and metabolic biomarkers, disruptions in the heme biosynthesis pathway are particularly informative. Porphyrins, the intermediates of this pathway, accumulate in urine when enzymatic steps are inhibited by heavy metals or oxidative stress. Evaluating specific urinary porphyrin fractions—including uroporphyrin (uP), heptacarboxyporphyrin (7cxP), hexacarboxyporphyrin (6cxP), pentacarboxyporphyrin (5cxP), and coproporphyrin (cP)—provides a biochemical window into toxicological effects and impaired detoxification in children with ASD. Figure 1 illustrates this biosynthetic sequence and demonstrates how environmental influences alter enzyme activity and porphyrin excretion profiles.

Research has connected mercury exposure in ASD to changed urine porphyrin profiles. Children with ASD have higher amounts of certain fractions, such as cP and 5cxP [12]. These abnormalities may be indirect indicators of systemic toxicity in addition to reflecting metabolic stress. Furthermore, children with ASD have shown signs of oxidative stress and reduced detoxification capacity, making porphyrin profiles a plausible marker of metabolic disturbance in this population [1,12].

Figure 1 illustrates the heme biosynthetic pathway, highlighting intermediates relevant for this study. Uroporphyrinogen III undergoes sequential decarboxylation by UROD to generate heptacarboxy-, hexa-, penta-, and coproporphyrinogen intermediates. These are oxidized to corresponding porphyrins—uP, 7cxP, 6cxP, 5cxP, cP—which are quantified in urine. Points of known heavy-metal inhibition are identified: lead strongly inhibits δ-aminolevulinic acid dehydratase (ALAD) and ferrochelatase (FECH), while mercury impairs coproporphyrinogen oxidase (CPOX). Oxidative stress, represented by lightning symbols, accelerates porphyrinogen oxidation, contributing to the altered urinary patterns seen in ASD.

Measurements of urinary porphyrin provide a non-invasive way to evaluate environmental and metabolic factors in ASD. Increased levels of coproporphyrins, uroporphyrins (uP), and heptacarboxyporphyrins (7cxP) have been linked to mercury exposure [11]. Furthermore, detoxification treatments have been shown to lower porphyrin levels in both human and animal models, indicating a clear link between exposure to environmental toxins and porphyrin metabolism [13]. Beyond porphyrins, other metabolic biomarkers have been associated with ASD, reflecting the heterogeneous and multifaceted nature of metabolic disturbances in this disorder. Recent studies have identified alterations in glycan metabolism and protein glycation as potential diagnostic markers. Validation studies using plasma protein glycation adducts—including advanced glycation end-products such as Nε-carboxymethyl-lysine (CML), Nω-carboxymethylarginine (CMA), and 3-deoxyglucosonederived hydroimidazolone (3DG-H)—have shown promise for ASD diagnosis in children aged 5–12 years, with diagnostic accuracy reaching 88% [14]. Genetic variants in glycosyltransferases and genes encoding glycosylated proteins have also been implicated in ASD susceptibility, suggesting that disruptions in brain glycosylation may contribute to neurodevelopmental abnormalities [15]. These findings underscore the importance of investigating multiple metabolic pathways and support a multivariate biomarker approach for understanding ASD pathophysiology.

This study aims to evaluate urinary porphyrin fractions and trace element concentrations in children with ASD compared to neurotypical controls, with stratification by age, sex, and autism severity as measured by the Childhood Autism Rating Scale (CARS). We hypothesize that children with ASD exhibit altered porphyrin profiles reflecting disrupted heme biosynthesis and that trace element imbalances—particularly elevated lead and altered Zn/Cu, Se/Pb, Cu/Se, and Zn/Pb ratios—contribute to oxidative stress and impaired detoxification. By integrating porphyrin and trace element analysis, we seek to identify non-invasive biomarkers for environmental and metabolic disturbances in ASD and to elucidate potential mechanistic links between heavy metal exposure, porphyrin metabolism, and neurodevelopmental outcomes.

## 2. Results

Urine samples from children with ASD and matched controls obtained, and the concentrations of total porphyrins and their specific fractions were examined. Urinary creatinine levels were established in order to normalize the data and reduce the effect of fluctuations in urine concentration. Because it lessens the variability brought about by variations in hydration status, this method is crucial for urine biomarker investigations [4].

### 2.1. Porphyrin Levels by Age

The study population was divided by age into three subgroups: ≤5 years; 5–10 years; and ≥10 years.

For subgroup ≤ 5, analysis of individual fractions and total porphyrins showed no significant differences between the ASD and control groups (*p* > 0.05) (Table 1). Results were lower for the ASD group in the case of total porphyrins and all individual fractions than for the control group (Table 1, Column ≤ 5).

For subgroup 5–10, analysis of individual fractions and total porphyrins also showed no significant differences between the ASD and control groups (*p* > 0.05) (Table 1). Results were higher for total porphyrins in the ASD group than in the control group, but the fractions uP, 7cxP, and cP were higher in the ASD group than in the control group, and 6cxP and 5cxP were lower (Table 1, Column 5–10). However, none of these findings reached statistical significance.

For the subgroup ≥ 10 years, the analysis of individual fractions and total porphyrins already showed a significant difference between the ASD and control groups (*p* ≤ 0.05) (Table 1). The results were higher for total porphyrins in the ASD group than in the control group. All fractions except 5cxP were significantly different, with only the 6cxP fraction being higher in the control group, and the 5cxP being lower (Table 1, Column 5–10).

Values are presented as means with 95% confidence intervals (CI) for descriptive purposes. Statistical comparisons were performed using Mann–Whitney U-test comparing medians, as data were not normally distributed (Shapiro–Wilk test). *p*-values represent non-parametric median comparisons. To facilitate interpretation of group differences, we calculated inter-group ratios (IGRs) by dividing the median porphyrin concentrations in ASD children by those in neurotypical controls (Figure 2). This metric visualizes relative increases or decreases across age groups and fractions. The IGRs show minimal differences in the youngest group (≤5 years), moderate elevation in most fractions among the middle group (5–10 years), and notable increases—particularly in uroporphyrin (1.27), 7cxP (1.32), coproporphyrins (1.54), and total porphyrins (1.42)—in children aged ≥10 years. An exception was 6cxP, which showed a decreased ratio (0.39), indicating a potential pathway-specific bottleneck in older ASD subjects. *p*-values for most porphyrins were highly significant (uroporphyrins *p* = 1.3 × 10^−6^, pentaporphyrins *p* = 9.21 × 10^−8^, total porphyrins *p* = 2.61 × 10^−5^) when comparing the youngest (≤5 years) and oldest (>10 years) age groups.

Bars represent the median concentration ratios of each porphyrin fraction in ASD children relative to neurotypical controls. Fractions include uroporphyrin (uP), heptacarboxyporphyrin (7cxP), hexacarboxyporphyrin (6cxP), pentacarboxyporphyrin (5cxP), coproporphyrins (cP), and total porphyrins. Age groups are categorized as ≤5 years, 5–10 years, and ≥10 years. The dashed line at IGR = 1 indicates no difference between groups.

A statistical study of the amounts of urine porphyrin in children with ASD among age groups reveals significant increases in porphyrins between the youngest and oldest age groups (≤5 vs. ≥10 years) (Table 2). *p*-values for most porphyrins were highly significant (uroporphyrins: *p* = 1.3 × 10^−6^, pentaporphyrins: *p* = 9.21 × 10^−8^, total porphyrins: *p* = 2.61 × 10^−5^). No significant changes in porphyrin fractions were found between the middle (5–10 years) and oldest (>10 years) age groups (all *p*-values > 0.1). Age-wise comparison also revealed moderate differences between the youngest (≤5 years) and middle (5–10 years) age groups. *p*-values were statistically significant for uroporphyrins (*p* = 0.032) and pentaporphyrins (*p* = 0.043), but not for other fractions. These findings suggest that some porphyrin fractions begin to accumulate earlier in life, likely reflecting gradual metabolic adaptations and potential environmental influences. Finally, no significant changes in porphyrin fractions were found between the middle (5–10 years) and oldest (≥10 years) age groups. All *p*-values were > 0.1, indicating no major differences in porphyrins after 5 years. Children with ASD had significantly lower levels of hexacarboxyporphyrin (6cxP/Creat) compared to controls (median 0.11 vs. 0.29 µmol/mol creatinine, *p* = 0.023). One explanation could be that detoxification pathways reach a functional threshold after a certain age, leading to stabilization.

### 2.2. Porphyrin Levels by Sex

Our results indicate that urinary porphyrin levels are consistently higher in children with ASD compared to healthy controls, with notable differences between boys and girls (Table 3). When stratified by sex, ASD males had significantly lower urinary hexacarboxyporphyrin (6cxP/Creat) compared to controls (median 0.10 vs. 0.25 µmol/mol creatinine; *p* = 0.04). In females, no significant differences were observed, although copro- and total porphyrin levels were moderately higher in ASD compared to controls. Uroporphyrin and heptacarboxyporphyrin were slightly elevated in ASD in both sexes, but without reaching statistical significance. These results suggest a sex-specific alteration in porphyrin metabolism, with ASD males showing a clear deficit in 6cxP.

A stepwise comparison was performed in order to more clearly understand the increase and evolution of distinct porphyrin percentages in ASD (Table 4). Children with ASD had significantly lower levels of hexacarboxyporphyrin (6cxP/Creat) compared to controls (median 0.11 vs. 0.29 µmol/mol creatinine; *p* = 0.023) (Table 4). Other porphyrin fractions showed non-significant differences, although consistent trends were observed (Table 4). Coproporphyrin (16.9 vs. 10.7 µmol/mol; *p* = 0.095), uroporphyrin (2.02 vs. 1.72 µmol/mol; *p* = 0.081), and total porphyrins (20.7 vs. 13.7 µmol/mol; *p* = 0.106) were higher in ASD, while heptacarboxyporphyrin also tended to be elevated (*p* = 0.088). These findings suggest that altered porphyrin metabolism in ASD is characterized by a reduction in 6-carboxyporphyrin together with trends toward accumulation of earlier porphyrin fractions, consistent with impaired decarboxylation processes.

Overall, these results indicate that the most consistent alteration in ASD is a significant decrease of hexacarboxyporphyrin, while other porphyrin fractions show upward trends without reaching statistical significance. The pattern of reduced 6cxP accompanied by moderate increases in earlier intermediates (uroporphyrin, heptacarboxyporphyrin, and coproporphyrin) suggests a potential bottleneck at the uroporphyrinogen decarboxylase step of heme biosynthesis. Importantly, the concordant elevation of coproporphyrin and total porphyrins, even if not statistically significant, points to a shifted porphyrin balance rather than a uniform increase across all fractions, supporting the idea that porphyrin disturbances in ASD are qualitative and pathway-specific.

### 2.3. Porphyrin Levels by CARS Score Categories

Because there was not a significant change between these groups, we compared the median porphyrin levels in children with ASD based on CARS scores (<37 vs. ≥37), which categorize children into less severe (median < 37) and more severe (median ≥ 37) ASD cases (Table 5). Table 5 presents the distribution of porphyrin fractions across ASD severity subgroups and controls. Significant elevations of copro- and pentacarboxyporphyrin were seen in ASD groups with moderate-to-severe symptoms, while hexacarboxyporphyrin was reduced. These findings suggest a severity-dependent shift in porphyrin metabolism.

We next stratified the ASD cohort in a binary fashion depending on whether CARS score median was <37 or ≥37 (Table 6). Our results show no significant differences in porphyrin fraction levels across ASD severity groups. The most notable (but still not significant) difference is in hexacarboxyporphyrins (6cxp, *p* = 0.091). We do observe a trend toward slightly higher porphyrin levels in the more severe group (median ≥ 37). Median values are marginally higher in the more severe group for uroporphyrins (uP), pentaporphyrins (5cxP), coproporphyrins (cP), and total porphyrins. These increases suggest a non-significant pattern that warrants further investigation, and they may indicate a trend toward greater metabolic disruption or environmental burden in children with more severe ASD.

The lack of significance in this analysis suggests that porphyrin levels alone may not be a strong biomarker for ASD severity, but they may still reflect underlying metabolic or detoxification abnormalities. The possible increased porphyrin fraction trend in the more severe ASD subgroup is a possible connection to higher oxidative stress and inflammation, increased environmental toxicant exposure, and dysregulated mitochondrial function, which can affect porphyrin metabolism.

### 2.4. Trace Elements Analysis and Their Ratios

Median urinary lead concentrations were significantly higher in children with ASD compared to controls (*p* = 0.004) (Table 7). No significant differences were observed in urinary concentrations of copper, zinc, or selenium between the ASD and control groups (*p* > 0.05 for all comparisons). The elevated Pb levels in ASD align with the observed increase in urinary porphyrin fractions such as coproporphyrin and pentacarboxyporphyrin, consistent with the inhibitory effect of Pb on porphyrin metabolism. Mercury (Hg) was detected in fewer samples and did not reach statistical significance but remains potentially biologically relevant (Table 7).

To explore the relative amounts and potential biological impact of trace elements, we calculated several ratios based on urinary concentrations. The Zn/Cu, Se/Pb, Cu/Se, and Zn/Pb ratios were significantly different between ASD and control groups (*p* < 0.001), with ASD children showing lower Zn/Cu and Se/Pb ratios and higher Cu/Se (Table 8). These findings suggest increased oxidative stress and reduced detoxification capacity in ASD. The Hg/Se ratio did not differ significantly between groups (*p* = 0.579), likely due to a smaller sample size and broader variability in Hg levels. These findings indicate that trace element ratios may better capture metabolic imbalances associated with ASD than individual elements alone.

## 3. Discussion

In the present work, we find that children with ASD had elevated amounts of total porphyrins and certain porphyrin fractions, such as uroporphyrins, heptaporphyrins, and coproporphyrins, compared to neurotypical controls. The significance of our findings was confirmed by statistical analyses utilizing the Mann–Whitney U-test for both the general study population and subgroups stratified by age. These results are consistent with earlier studies showing that ASD is associated with changes in detoxification pathways or heme production [18].

Notably, children older than 10 years showed greater increases in porphyrin fractions. Interestingly, hexacarboxyporphyrin (6cxP) was significantly reduced in older children with ASD, a finding that diverges from the trend seen in other fractions. This may point to a bottleneck or enzymatic block within the heme biosynthesis pathway, possibly affecting the decarboxylation from hexacarboxyporphyrin to pentacarboxyporphyrin. The need for further investigation of environmental and metabolic factors as possible causes of the observed disparities is highlighted by the increasing urinary levels of porphyrin fractions, especially in older children [27,28].

Such pathway-specific anomalies have been observed in metabolic studies of ASD [9,10]. This pattern raises the possibility that these disruptions are caused by age-related metabolic changes or cumulative exposure to environmental agents such as heavy metals [29,30]. Porphyrin metabolism is known to be hampered by heavy metals, such as lead and mercury, and the buildup of these elements has been connected to metabolic disorders associated with ASD [11,29]. Although elevated porphyrin levels may suggest a role for environmental exposures, no direct measurements were made. In general, our results suggest a potential association between disrupted heme biosynthesis and systemic stressors in children with ASD, but causality cannot be inferred from this cross-sectional dataset.

Our observed absence of significant porphyrin differences in children under 10 may reflect developmental immaturity of detoxification pathways, shorter cumulative exposure to environmental agents, or lower systemic oxidative stress. These factors can delay the onset of detectable metabolic disruptions until later childhood [1,13].

Our findings of elevated porphyrin fractions—particularly in older children with ASD—are consistent with a growing body of literature highlighting abnormal porphyrin metabolism in autism. For instance, Khaled et al. reported significantly increased urinary porphyrins (uroporphyrin, coproporphyrin) in Egyptian children with ASD compared to controls, which they interpreted as a sign of possible heavy metal exposure, particularly mercury [30]. Similarly, Nataf et al. (2006) and Geier et al. (2006) identified porphyrin abnormalities in ASD cohorts and linked them to environmental toxicity and detoxification dysfunction [9,12]. More recently, Indika et al. (2024) used a metabolomic approach and confirmed widespread disruptions in porphyrin metabolism in children with ASD, emphasizing systemic oxidative and mitochondrial stress and proposing potential therapeutic interventions [10]. In addition, Bjørklund et al. (2018) reviewed multiple biomarker studies and emphasized porphyrin alterations as potential early indicators of metabolic dysregulation, especially when paired with oxidative stress and redox imbalance markers [4]. These observations align with our results in older children, who may have experienced prolonged environmental stress or accumulated toxin burden. The age-dependent increase in porphyrin fractions, particularly pronounced between ages ≤5 and >10 years, likely reflects cumulative environmental exposures to heavy metals, progressive oxidative stress, and age-related metabolic shifts in heme biosynthesis. This pattern suggests that metabolic disturbances associated with ASD become more biochemically evident over time, potentially due to sustained toxic exposure or declining detoxification capacity. The stabilization of porphyrin levels between middle childhood (5–10 years) and later childhood (>10 years) may indicate that detoxification pathways reach a functional threshold or adaptive response after prolonged exposure. Alternatively, developmental changes in enzyme expression and metabolic regulation during this period may modulate porphyrin excretion patterns. The specific reduction in hexacarboxyporphyrin (6cxP) accompanied by elevations in earlier pathway intermediates (uroporphyrin, heptacarboxyporphyrin, coproporphyrin) suggests a bottleneck at the uroporphyrinogen decarboxylase step. This enzyme catalyzes the sequential removal of carboxyl groups from uroporphyrinogen to coproporphyrinogen and may be particularly vulnerable to heavy metal inhibition or oxidative damage. The disrupted ratio of porphyrin fractions, rather than uniform elevation, supports pathway-specific dysregulation in ASD.

Interestingly, while most studies focused on uroporphyrins and coproporphyrins, we also observed significant changes in heptaporphyrins and total porphyrins, which may reflect upstream enzymatic disturbances (e.g., uroporphyrinogen III decarboxylase or coproporphyrinogen oxidase) and age-dependent metabolic responses. The lack of significant findings in younger children could be due to immature detoxification systems or reduced exposure duration—points raised in prior toxicological studies [31].

Recent studies have highlighted the potential role of environmental toxicants in the etiology of ASD. Studies indicate that children with ASD have significantly increased levels of urinary porphyrins associated with mercury toxicity compared to neurotypical controls [32]. Various environmental toxicants, including heavy metals, persistent organic pollutants, and emerging chemicals of concern, have been linked to ASD development [33]. A systematic review found that 92% of studies examining estimated toxicant exposures reported an association with ASD risk [34]. The review also identified genetic polymorphisms that may increase susceptibility to toxicants in individuals with ASD. However, further high-quality epidemiological research is needed to confirm these findings and elucidate the complex interactions between genetic factors and environmental toxicants in ASD etiology.

Recent research has also explored the potential of urinary porphyrins as biomarkers for ASD [35]. Some studies have reported elevated porphyrin levels in children with ASD compared to controls, with correlations between porphyrin levels and ASD severity [36,37]. However, the validity of porphyrins as a diagnostic tool remains controversial. Shandley et al. [38] found no evidence to support the use of porphyrin levels to diagnose ASD or predict severity. The heterogeneity of ASD underscores the need for reliable biomarkers to identify subtypes and inform interventions [37]. Vargason et al. suggest that a multivariate approach, combining behavioral, genetic, and metabolic markers, may be more effective for ASD diagnosis than single biomarkers [39].

This study’s second part examined the association between porphyrin concentrations and the severity of ASD as measured by the Childhood Autism Rating Scale (CARS). The purpose of this analysis was to ascertain whether porphyrin levels may be used as biomarkers to monitor clinical severity or developmental delays. Porphyrin concentrations in the ASD and control groups did not, however, differ significantly. Furthermore, no relationship between porphyrin levels and CARS scores was discovered, indicating that although porphyrins might reflect systemic metabolic stress, their levels do not correlate with autism severity [29,30].

A further major aspect of our investigation was the evaluation of urinary trace elements. We found significantly elevated urinary lead (Pb) concentrations in the ASD group (*p* = 0.004), while copper (Cu), zinc (Zn), and selenium (Se) did not differ significantly between groups. Lead is a well-known inhibitor of key enzymes in the heme biosynthetic pathway, such as δ-aminolevulinic acid dehydratase (ALAD) and ferrochelatase, which can result in the accumulation of specific porphyrin intermediates, including cP and 5cxP. These were the same fractions elevated in our study in the ASD group. This enzymatic interference provides a plausible mechanistic explanation for the observed metabolic changes in ASD and underscores Pb’s role in contributing to systemic biochemical disturbances.

In a smaller subset of participants, urinary mercury (Hg) levels were also assessed. Although no statistically significant differences were observed between the ASD and control groups, Hg remains a heavy metal of concern due to its established neurotoxic potential and its capacity to alter porphyrin metabolism. Previous studies have suggested that even low-level mercury exposure may be associated with atypical porphyrin patterns, such as increases in precoproporphyrin (prcP) or pentacarboxyporphyrin, which are consistent with some patterns observed in ASD cohorts. However, the limited sample size in our Hg subgroup precludes strong conclusions, and further investigation with larger datasets is warranted.

The absence of significant group differences for other elements (Cu, Zn, Se) suggests a more specific involvement of Pb—and possibly Hg—in modulating porphyrin metabolism. While selenium and zinc have antioxidant properties and may modulate heavy metal toxicity, their urinary levels were not markedly different in our ASD cohort.

Importantly, trace element ratios may serve as more reliable indicators of metabolic dysfunction than absolute element levels. The ratios reflect biological interactions—such as antagonism between Se and Pb or Zn and Cu—that influence detoxification, oxidative stress, and immune function. Analysis of trace element ratios revealed significant alterations in ASD. The Zn/Cu ratio was lower, consistent with increased oxidative stress and impaired antioxidant defense. The Se/Pb ratio was also reduced, suggesting diminished detoxification capacity against Pb. The Cu/Se ratio was elevated, reinforcing redox imbalance. Finally, Zn/Pb was lower, pointing toward a higher Pb burden relative to Zn protective capacity. These ratio shifts complement porphyrin findings, indicating that heavy metal exposure and redox imbalance jointly contribute to the biochemical profile of ASD.

Prior studies on trace elements in ASD have reported significantly lower zinc (Zn) and selenium (Se) levels in ASD children compared to controls [16,40,41]. Conversely, copper (Cu) and lead (Pb) levels were often higher in ASD groups [16,40]. The Zn/Cu ratio was significantly lower in ASD patients and negatively correlated with ASD severity [23]. These imbalances may contribute to impaired antioxidant defense and reduced detoxification potential in ASD [16,40]. Notably, Se levels showed a negative correlation with total ADOS scores in one study [41]. The Zn/Cu ratio has been proposed as a potential biomarker for ASD, with one study reporting a sensitivity of 90.0% and specificity of 91.7% at an optimal cut-off value of 0.665 [23]. These findings support the view that functional trace element imbalances may exacerbate or reflect the metabolic disruptions underlying porphyrin abnormalities in ASD.

Genetic polymorphisms in genes encoding metal metabolism and detoxification enzymes may modulate individual susceptibility to heavy metal-induced neurotoxicity and porphyrin pathway disruption. Key candidate genes include glutathione S-transferase (GST) family members, particularly GSTP1, GSTM1, and GSTT1, which influence lead detoxification capacity through conjugation reactions. Variants in these genes have been associated with elevated blood lead concentrations and differential neurodevelopmental outcomes in exposed children [42]. Metallothionein genes (MT1A, MT2A) encode cysteine-rich proteins that sequester and detoxify heavy metals including lead, mercury, cadmium, and zinc. Genetic variation in metallothionein expression may affect metal accumulation and oxidative stress burden [43]. ATP-binding cassette (ABC) transporters, including ABCB1 (P-glycoprotein) and ABCC1 (MRP1), regulate cellular efflux of metal ions and their conjugates, thereby influencing tissue metal distribution and toxicity [44]. Additionally, enzymes in the porphyrin biosynthesis pathway itself—particularly δ-aminolevulinic acid dehydratase (ALAD) and ferrochelatase (FECH)—contain lead-sensitive zinc cofactors or iron-binding sites, making them vulnerable to metal-induced inhibition. The ALAD gene has a common polymorphism (ALAD1/ALAD2) that affects lead binding affinity and has been linked to differential neurotoxic effects in exposed populations [15]. The interaction between genetic susceptibility variants and environmental metal exposure likely contributes to the heterogeneity observed in ASD and may explain why some children develop more pronounced porphyrin abnormalities despite similar environmental conditions. Future studies incorporating genetic profiling alongside metabolic biomarker assessment could identify high risk subgroups and inform personalized intervention strategies. Our results demonstrate the complicated nature of metabolic problems linked to ASD. Although elevated levels of porphyrin might be a sign of environmental exposures, poor detoxification, or systemic oxidative stress, their function as biomarkers for diagnosis or severity tracking is yet unknown [4,45]. To further understand the use of porphyrins in ASD research, future studies should integrate genetic predispositions, environmental exposure data, and broader metabolic profiling [11,18].

### 3.1. Strengths of the Study

Urinary creatinine was used for normalization, ensuring that fluctuations in urine concentration would not confound the observed differences in porphyrin levels. The study found age-dependent patterns in porphyrin metabolism by examining both the total population and certain age groups. The results support the body of research on heavy metal toxicity and offer a foundation for investigating environmental factors that may contribute to ASD. The Childhood Autism Rating Scale (CARS) offered a standardized way to link developmental deficits with biochemical abnormalities. An important strength of this study is the simultaneous evaluation of metabolic intermediates and trace environmental toxicants in a non-invasive biological matrix. This integrative approach allows for a broader understanding of the biochemical environment in ASD and offers insights into how environmental exposures may influence internal metabolic pathways.

### 3.2. Weaknesses of the Study

This cross-sectional design precludes causal inference. The absence of direct measurements of environmental toxicants, such as methylmercury or lead, constrains attribution of observed associations to specific exposures. Imperfect age- and sex-matching between ASD and control groups, along with unmeasured confounders—including diet, medication use, and genetic variability in metal metabolism—may have introduced bias.

Despite subgroup analysis by sex and age, the overall sex imbalance in the study population (ASD: 78% male; controls: 53% male) may influence the interpretation of some findings. Previous studies have reported sex-specific patterns in porphyrin metabolism in ASD [4,30], which may contribute to differential vulnerability or exposure.

Small subsample sizes for certain trace element analyses, the lack of an a priori power calculation, and the absence of formal correction for multiple comparisons further limit statistical robustness. Consequently, these findings should be considered exploratory and hypothesis-generating, underscoring the need for larger, well-controlled studies incorporating direct toxicant assessment and additional oxidative stress biomarkers.

## 4. Materials and Methods

### 4.1. Characteristics of Study Participants

The study population consisted of 241 children, 165 with ASD (128 boys/36 girls) and 76 healthy controls (40 boys/36 girls). In the ASD group, the subjects’ average age was 11 years, with a range of 5–17 years. The control group included neurotypical children without any acute or chronic illness who were, on average, 9 years of age, with a range of 1–17 years. For the ASD group, inclusion criteria were: (1) children aged 5–17 years; (2) confirmed diagnosis of ASD according to DSM-5 criteria by a multidisciplinary team; (3) completed CARS assessment. For the control group, inclusion criteria were: (1) children aged 1–17 years; (2) neurotypical development without known neurological or psychiatric conditions; and (3) no acute or chronic illness at time of sample collection. Exclusion criteria for both groups included: (1) current supplementation with vitamins or magnesium; (2) acute or chronic medical illness (including infections, metabolic disorders, or inflammatory conditions); and (3) for controls, any known genetic, neurological, or psychiatric disorder. These criteria were applied to ensure comparability between groups and minimize confounding from supplementation or acute illness. Additional behavioral ratings were based on a standardized classification of behavior for children with ASD developed by the local educational authority for providing additional school support [46,47,48]. All ASD diagnoses were confirmed through clinical observation and standardized behavioral testing. The Childhood Autism Rating Scale (CARS) was applied for severity classification [49]. While the CARS has limitations, it is commonly used in clinical and research contexts, particularly where more resource-intensive diagnostic tools like ADOS are unavailable. The CARS questionnaire was filled out by all parents in conjunction with a psychologist. Control children were age- and sex-matched neurotypical individuals without known neurological or psychiatric conditions. The interpretation of CARS scores in our study was as follows:

Score below 28: Typically, a score below 28 indicates that the child is not likely to have ASD. However, it is important to consider other factors and use clinical judgment in making a final diagnosis.

Score between 28 and 36: This range suggests the possibility of mild to moderate autism symptoms. Further evaluation and observation may be needed to determine if the child meets the criteria for ASD.

Score above 36.5: Scores above 36.5 indicate a higher likelihood of significant autism symptoms. It suggests a stronger possibility of the child meeting the diagnostic criteria for ASD.

The study was approved by the National Medical Ethics Committee of Slovenia (Ref. No. 0120-201/2016/6 from 3 February 2021), and informed consent was obtained from parents or guardians.

### 4.2. Urine Sample Collection and Analysis

Urine samples were collected in sterile containers and stored at −80 °C until analysis. Urinary porphyrins were quantified using high-performance liquid chromatography (HPLC) based on the method described by Armbruster et al. (1983) [50]. Separation was performed on an RP18 HPLC column (5 μm, 4.6 mm × 250 mm) with a mobile phase consisting of acetonitrile and acetate buffer (pH 4.5). Detection was performed using a Shimadzu RF-20A fluorescence detector, with excitation and emission wavelengths set to 405 nm and 620 nm, respectively. Results were quantified against calibration curves prepared with known standards of porphyrins.

Using inductively coupled plasma mass spectrometry (ICP-MS) (7700x, Agilent Technologies, Santa Clara, CA, USA), urinary concentrations of specific trace elements (Pb, Cu, Zn, Se, and Hg) were measured in accordance with normal sample preparation procedures as previously described [51]. Every result was represented as µg/g creatinine after being normalized to urine creatinine. Creatinine normalization is the standard method for urinary biomarker analysis because creatinine excretion is relatively constant (approximately 1 g/day in adults, proportional to muscle mass in children) and independent of urine volume or hydration status. This normalization accounts for variations in urine concentration and dilution, allowing valid comparison of analyte concentrations between samples collected at different times and under different hydration conditions. Without creatinine normalization, differences in urinary analyte concentrations might reflect merely differences in urine dilution rather than true physiological changes.

Urinary creatinine levels were determined using the Jaffé reaction. This involves a colorimetric assay where creatinine reacts with picric acid under alkaline conditions to form a red–orange complex, measured at 520 nm. Porphyrin concentrations were normalized to urinary creatinine to account for variations in urine concentration.

### 4.3. Porphyrin Nomenclature

Porphyrins are named according to the number of carboxyl groups attached to the tetrapyrrole ring structure. In the heme biosynthesis pathway, sequential decarboxylation removes carboxyl groups, proceeding from uroporphyrinogen (8 carboxyl groups) to coproporphyrinogen (4 carboxyl groups). The porphyrin fractions measured in this study are:-Uroporphyrin (uP): 8-carboxyl porphyrin, the earliest intermediate.-Heptacarboxyporphyrin (7cxP): 7-carboxyl porphyrin.-Hexacarboxyporphyrin (6cxP): 6-carboxyl porphyrin.-Pentacarboxyporphyrin (5cxP): 5-carboxyl porphyrin.-Coproporphyrin (cP): 4-carboxyl porphyrin, the end product of decarboxylation.

These fractions accumulate and are excreted in urine when the biosynthetic pathway is disrupted by heavy metals, oxidative stress, or enzyme deficiencies. The pattern of accumulated fractions provides diagnostic information about the site and nature of pathway disruption.

### 4.4. Statistical Analysis

All statistical analyses were performed using MedCalc Statistical Software version 20.011 (MedCalc Software Ltd., Ostend, Belgium). The data were first assessed for distribution using the Shapiro–Wilk test. As most variables did not follow a normal distribution, non-parametric methods were used throughout the analysis.

Group comparisons between children with ASD and healthy controls were performed using the Mann–Whitney U-test, which is appropriate for comparing medians between two independent groups. This test was also used for subgroup comparisons based on sex and age categories.

To explore age-related trends, children were stratified into three age groups: ≤5 years, 5–10 years, and ≥10 years. Pairwise comparisons among these groups were conducted using non-parametric tests of equality of medians.

The relationship between porphyrin levels and autism severity, based on Childhood Autism Rating Scale (CARS) scores, was examined by dividing the ASD group into two subgroups: those with CARS < 37 (mild to moderate ASD) and CARS ≥ 37 (severe ASD). Median porphyrin concentrations were compared between these groups using the median equality test.

Descriptive versus Inferential Statistics: Table 1 presents mean concentrations with 95% confidence intervals to provide comprehensive descriptive information about central tendency and variability for each group. However, because the Shapiro–Wilk test revealed that most variables did not follow normal distribution, non-parametric Mann–Whitney U-test based on medians was used for all statistical comparisons. This dual approach follows statistical best practice: means and confidence intervals offer intuitive descriptive summaries, while median-based non-parametric tests provide valid hypothesis testing when distributional assumptions for parametric tests are violated. All *p*-values reported in tables represent results of non-parametric tests comparing medians, not means.

A *p*-value of <0.05 was considered statistically significant. Results were presented as median values (μmol/mol creatinine) with corresponding *p*-values to indicate statistical significance.

## 5. Conclusions

Our dataset provides useful data about porphyrin metabolism in ASD, highlighting the importance of age and possible environmental factors. Elevations in urinary porphyrin fractions were statistically significant only in the oldest age group (≥10 years), suggesting age-related biochemical divergence in ASD. Our findings suggest that urinary porphyrins may reflect systemic metabolic disturbances in a subset of children with ASD, particularly older children. However, these results are exploratory and should be interpreted with caution. Future prospective studies are needed to confirm these associations and clarify their clinical utility. The absence of connection with autism severity via CARS implies that porphyrins may reflect systemic metabolic disturbances rather than act as direct indicators of autism severity, even if significant differences were observed in older children. Urinary porphyrin profiling, combined with trace element analysis and calculation of functional ratios, offers a promising approach to assessing environmental and metabolic dysregulation in children with ASD. Elevated Pb levels and altered Zn/Cu, Se/Pb, Cu/Se, and Zn/Pb ratios suggest increased oxidative stress and impaired detoxification. These biomarkers may support early detection, risk stratification, and targeted interventions.

## Figures and Tables

**Figure 1 ijms-26-10478-f001:**
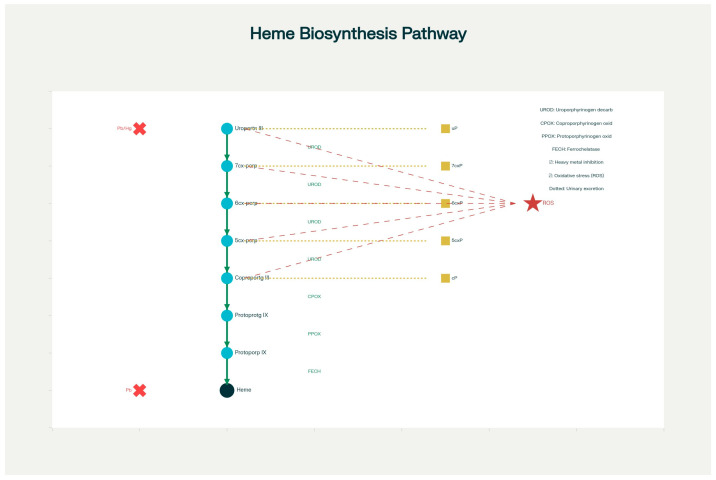
Heme biosynthesis pathway showing porphyrin intermediates, enzymes, and disruptions by heavy metals and oxidative stress. The sequential conversion from uroporphyrinogen III to coproporphyrinogen III, protoporphyrinogen IX, and heme is mediated by UROD, CPOX, PPOX, and FECH enzymes (solid arrows). Red “X” markers indicate inhibitory effects of lead (Pb) and mercury (Hg) on specific enzymatic steps. Lightning symbols denote oxidative stress (ROS)–induced oxidation of porphyrinogens to urinary porphyrins. Dashed arrows represent the oxidation and urinary excretion of porphyrin intermediates, which are measured as biomarkers in this study. Measured urinary porphyrins (uP, 7cxP, 6cxP, 5cxP, and cP) are color-coded, representing the stepwise decarboxylation of the heme synthesis pathway relevant to ASD biomarker analysis.

**Figure 2 ijms-26-10478-f002:**
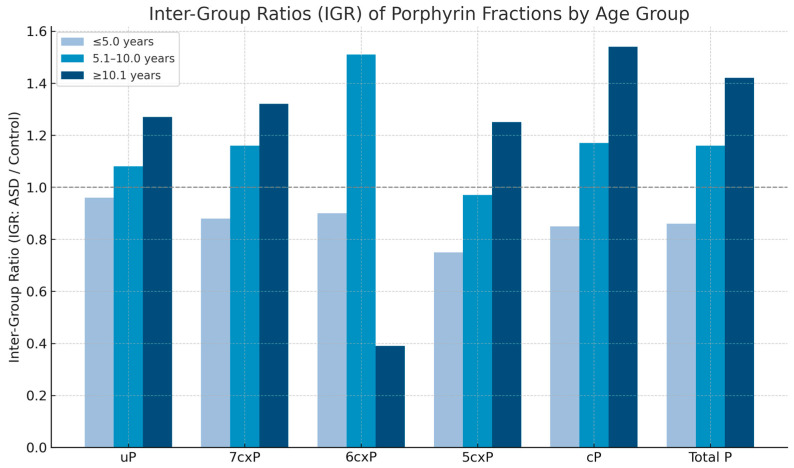
Inter-Group Ratios (IGR = ASD/control) of urinary porphyrin fractions by age group.

**Table 1 ijms-26-10478-t001:** Mean concentrations with 95% CI of individual porphyrin fractions expressed per creatinine as a function of age, with significance assessed by *t*-test.

Porphyrin Fraction	≤5			5–10			≥10		
	Control Group	ASD Group	*p*-Value	Control Group	ASD Group	*p*-Value	Control Group	ASD Group	*p*-Value
**uP/creat (µmol/mol)**	2.66 (1.87–3.44)	2.55 (2.10–2.99)	0.73	1.94 (1.68–2.19)	2.10 (1.82–2.36)	0.86	1.56 (1.31–1.79)	1.98 (1.77–2.19)	0.007
**7cxP/creat (µmol/mol)**	0.58 (0.37–0.78)	0.51 (0.38–0.63)	0.54	0.37 (0.30–0.42)	0.43 (0.25–0.50)	0.67	0.34 (0.27–0.40)	0.45 (0.38–0.50)	0.006
**6cxP/creat (µmol/mol)**	1.24 (0.07–2.55)	1.1 (0.05–2.18)2	0.24	0.41 (0.19–0.62)	0.62 (0.29–0.94)	0.88	0.89 (0.36–1.40)	0.35 (0.21–0.49)	0.01
**5cxP/creat (µmol/mol)**	0.59 (0.32–0.85)	0.44 (0.32–0.55)	0.39	0.36 (0.26–0.45)6	0.35 (0.27–0.42)	0.49	0.24 (0.17–0.30)	0.30 (0.19–0.40)	0.30
**cP/creat (µmol/mol)**	29.63 (16.11–43.14)	25.30 (18.34–32.25)	0.54	15.45 (11.69–19.21)	18.01 (14.99–21.09)	0.68	10.23 (7.63–12.82)	15.78 (12.87–18.67)	0.02
**Total P/creat (µmol/mol)**	34.94 (20.19–49.69)	30.03 (22.02–38.04)	0.52	18.57 (14.50–22.61)	21.59 (18.10–25.06)	0.56	13.36 (10.07–16.64)	18.93 (15.80–22.04)	0.03

**Table 2 ijms-26-10478-t002:** Analysis of the age-related increase in individual porphyrin fractions.

Age Group Comparison	uP (*p*-Value)	7cxP (*p*-Value)	6cxP (*p*-Value)	5cxP (*p*-Value)	cP (*p*-Value)	Tot. (*p*-Value)
≤5 vs. 5–10	**0.032**	0.141	0.549	**0.043**	0.089	0.088
5–10 vs. ≥10	0.741	0.293	0.174	0.180496	0.287	0.296
≤5 vs. ≥10	**1.3 × 10^−6^**	**0.012**	**0.027**	**9.21 × 10^−8^**	**4.3 × 10^−5^**	**2.61 × 10^−5^**

**Table 3 ijms-26-10478-t003:** Median concentrations of individual porphyrin fractions expressed per creatinine as a function of the sex of the child.

Sex	uP/Creat		*p*	7cxP/Creat		*p*	6cxP/Creat		*p*	5cxP/Creat		*p*	cP/Creat		*p*	Total P/Creat		*p*
	C	ASD		C	ASD		C	ASD		C	ASD		C	ASD		C	ASD	
F	2.00 (1.45–2.41)	2.31 (1.42–2.92)	0.33	0.41 (0.27–0.51)	0.49 (0.27–0.58)	0.33	0.40 (0.08–1.11)	0.23 (0.08–0.94)	0.79	0.33 (0.28–0.41)	0.34 (0.26–0.48)	0.64	15.06 (6.05–21.53)	19.30 (6.49–24.21)	0.17	18.90 (8.30–25.15)	23.38 (9.01–31.51)	0.19
M	1.76 (1.02–2.35)	1.94 (1.31–2.57)	0.11	0.37 (0.21–0.48)	0.44 (0.25–0.58)	0.14	0.25 (0.07–0.61)	0.10 (0.04–0.37)	**0.04**	0.35 (0.20–0.42)	0.35 (0.19–0.49)	0.93	15.37 (4.18–24.95)	17.97 (6.26–26.24)	0.38	18.32 (5.86–29.91)	21.32 (8.54–30.58)	0.37
*p*	0.165	0.269		0.298	0.455		0.337	**0.009**		0.838	0.312		0.874	0.660		0.980	0.568	

Values are presented as medians with interquartile ranges (IQR), normalized to urinary creatinine (µmol/mol). Statistical comparisons were made using the Mann–Whitney U test. *p* = between sex; *p* = between control group (C) and ASD group.

**Table 4 ijms-26-10478-t004:** Urinary porphyrin fractions in ASD and control groups.

	Control	ASD	
	Median (IQR)		*p*
**5cxP/Creat**	0.24 (0.13 to 0.47)	0.26 (0.14 to 0.41)	0.870
**6cxP/Creat**	0.29 (0.07 to 0.75)	0.11 (0.04 to 0.44)	**0.023**
**7cxP/Creat**	0.35 (0.24 to 0.49)	0.39 (0.26 to 0.58)	0.088
**cP/Creat**	10.70 (5.38 to 21.84)	16.91 (6.47 to 26.07)	0.095
**uP/Creat**	1.72 (1.23 to 2.35)	2.02 (1.34 to 2.67)	0.081
**Total P/Creat**	13.71 (8.03 to 25.97)	20.69 (8.78 to 30.79)	0.106

Values are expressed as medians with interquartile ranges (IQR), normalized to urinary creatinine (µmol/mol). Statistical comparisons were performed using the Mann–Whitney U test.

**Table 5 ijms-26-10478-t005:** Comparison of ASD and control porphyrin fraction median concentrations with IQR stratified by autism severity.

CARS Score	≤28			29–36.5			≥37		
Porphyrin Fraction	Control Group	ASD Group	*p*-Value	Control Group	ASD Group	*p*-Value	Control Group	ASD Group	*p*-Value
**uP/creat (µmol/mol)**	1.65 (1.20–2.49)	1.85 (1.40–2.65)	0.311	1.65 (1.20–2.49)	1.69 (1.26–2.47)	0.662	1.65 (1.20–2.49)	1.95 (1.29–2.52)	0.086
**7cxP/creat (µmol/mol)**	0.36 (0.25–0.48)	0.39 (0.27–0.58)	0.198	0.36 (0.25–0.48)	0.30 (0.20–0.45)	0.245	0.36 (0.25–0.48)	0.38 (0.27–0.66)	0.640
**6cxP/creat (µmol/mol)**	0.22 (0.06–0.65)	0.09 (0.03–0.46)	0.198	0.22 (0.06–0.65)	0.09 (0.03–0.47)	0.662	0.22 (0.06–0.65)	0.24 (0.04–0.53)	0.877
**5cxP/creat (µmol/mol)**	0.22 (0.11–0.42)	0.22 (0.15–0.40)	0.812	0.22 (0.11–0.42)	0.23 (0.10–0.40)	0.760	0.22 (0.11–0.42)	0.25 (0.13–0.42)	0.559
**cP/creat (µmol/mol)**	10.97 (5.30–21.75)	12.27 (6.10–26.78)	0.668	10.97 (5.30–21.75)	12.37 (4.83–21.55)	1	10.97 (5.30–21.75)	14.63 (4.16–26.63)	0.391
**Total P/creat (µmol/mol)**	14.20 (7.64–25.68)	15.58 (8.34–30.96)	1	14.20 (7.64–25.68)	14.72 (7.02–25.94)	1	14.20 (7.64–25.68)	17.71 (7.16–31.39)	0.784

Values are medians (IQR). Statistical test: Mann–Whitney U-test.

**Table 6 ijms-26-10478-t006:** Test of equality of porphyrin concentration medians between ASD and control by level of autism severity.

	uP	7cxP	6cxP	5cxP	cP	Total P
**Median < 37**	1.79	0.36	0.09	0.22	12.27	15.58
**Median ≥ 37**	1.95	0.36	0.22	0.25	13.67	17.28
**Median (Total)**	1.87	0.36	0.12	0.22	13.28	16.84
***p*-value**	0.205	0.774	0.091	0.571	0.673	0.673

**Table 7 ijms-26-10478-t007:** Urinary concentrations of selected trace elements (µg/g creatinine) in children with ASD and neurotypical controls. Statistical comparisons were performed using the Mann–Whitney U-test.

Element	ASD Group (Median [IQR])	Control Group (Median [IQR])	*p*-Value
**Lead (Pb)**	1.275 [0.865–1.631]	0.822 [0.588–1.377]	0.004
**Copper (Cu)**	10.234 [7.745–11.989]	9.290 [5.941–13.844]	0.157
**Zinc (Zn)**	464.657 [188.072–698.387]	455.894 [321.363–597.039]	0.273
**Selenium (Se)**	21.891 [12.999–31.624]	22.974 [18.142–31.495]	0.343
**Mercury (Hg)**	0.13 [0.0725–0.296]	0.237 [0.143–0.414]	0.329

**Table 8 ijms-26-10478-t008:** Median trace element ratios in urine samples from ASD and control groups.

Ratio	ASD (Median)	Control (Median)	*p*-Value	Interpretation
**Zn/Cu**	45.4	49.1	<0.001	Lower in ASD—suggests increased oxidative stress [16,17,18]
**Se/Pb**	17.2	27.9	<0.001	Lower in ASD—suggests impaired Pb detox [19,20,21]
**Cu/Se**	0.47	0.40	<0.001	Higher in ASD—reflects pro-oxidant shift [18,22,23]
**Hg/Se**	0.01	0.01	0.579	No significant difference [20,24,25]
**Zn/Pb**	364.4	554.6	<0.001	Lower in ASD—further support for Pb burden [18,19,26]

Values are expressed as medians with units normalized to urinary creatinine (µg/g creatinine for elements). Statistical significance was determined using the Mann–Whitney U-test.

## Data Availability

The data that support the findings of this study are available from the study’s principal investigator—O.J.—upon reasonable request.

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
