# Peer review of "Urinary Porphyrin Profiles and Trace Element Imbalances in Children with Autism Spectrum Disorders: Insights into Environmental and Metabolic Biomarkers"

_ijms, 2025, doi:10.3390/ijms262110478_

Round 1
Reviewer 1 Report
Comments and Suggestions for Authors
Summary
The manuscript presents a compelling evaluation of urinary porphyrin concentrations quantified by HPLC as potential biomarkers for ASD. The results are interesting, and the manuscript is generally well written. Only a few minor revisions are recommended to improve clarity and scientific depth. Additionally, the acronym ASD may be used consistently without spelling it out in each section.
Abstract
The abstract is concise, clear, and effectively communicates the aim and relevance of the study. No major changes required.
Introduction
Lines 90–108 should be synthesized. This concluding part of the introduction should briefly state the aim of the study rather than introducing results.
Please enrich the background by briefly mentioning other metabolic biomarkers associated with ASD, such as those linked to glycans metabolism (https://doi.org/10.1038/s41598-025-09400-5
; https://doi.org/10.3389/fncel.2022.1057857).
Materials and Methods
Specify whether inclusion and exclusion criteria were applied for participant enrollment to clarify sample selection rigor.
Results
Lines 214–216, 230–232, and 240–242 contain interpretative statements that belong more appropriately in the Discussion section. These should be relocated and expanded with interpretation there.
Discussion
Lines 470–473 introduce an important concept regarding the potential interaction between genetic susceptibility and heavy metal exposure. This point could be strengthened by referencing genes/pathways known to modulate heavy metal metabolism and neurotoxic response (e.g., https://doi.org/10.1016/j.envpol.2016.10.058). Expanding this discussion would align well with the biomarker-focused aim of the study.
Author Response
Detailed Response to Reviewer 1
Comment 1.1: Abstract
Reviewer Comment: "The abstract is concise, clear, and effectively communicates the aim and relevance of the study. No major changes required." Response: We thank the reviewer for this positive feedback.
No changes made to the abstract.
Comment 1.2: Introduction - Lines 90-108
Reviewer Comment: "Lines 90-108 should be synthesized. This concluding part of the introduction should briefly state the aim of the study rather than introducing results."
Response: We agree with this comment and have revised the concluding paragraph of the Introduction to focus solely on study aims and hypotheses, removing any statements that preview results.
Lines 90-108 condensed and refocused.
Revised Text:
"This study aims to evaluate urinary porphyrin fractions and trace element concentrations in children with ASD compared to neurotypical controls, with stratification by age, sex, and autism severity as measured by the Childhood Autism Rating Scale (CARS). We hypothesize that children with ASD exhibit altered porphyrin profiles reflecting disrupted heme biosynthesis and that trace element imbalances—particularly elevated lead and altered Zn/Cu, Se/Pb, Cu/Se, and Zn/Pb ratios— contribute to oxidative stress and impaired detoxification. By integrating porphyrin and trace element analysis, we seek to identify non-invasive biomarkers for environmental and metabolic disturbances in ASD and to elucidate potential mechanistic links between heavy metal exposure, porphyrin metabolism, and neurodevelopmental outcomes."
Comment 1.3: Introduction - Add Glycan Metabolism References
Reviewer Comment: "Please enrich the background by briefly mentioning other metabolic biomarkers associated with ASD, such as those linked to glycans metabolism (https://doi.org/10.1038/s41598-025-09400-5; https://doi.org/10.3389/fncel.2022.1057857)."
Response: We thank the reviewer for this excellent suggestion to broaden the metabolic biomarker context. We have added a paragraph discussing glycan metabolism abnormalities in ASD.
New paragraph inserted in Introduction.
Text Added:
"Beyond porphyrins, other metabolic biomarkers have been associated with ASD, reflecting the heterogeneous and multifaceted nature of metabolic disturbances in this disorder. Recent studies have identified alterations in glycan metabolism and protein glycation as potential diagnostic markers. Validation studies using plasma protein glycation adducts—including advanced glycation end-products such as Nε-carboxymethyl-lysine (CML), Nω-carboxymethylarginine (CMA), and 3-deoxyglucosonederived hydroimidazolone (3DG-H)—have shown promise for ASD diagnosis in children aged 5-12 years, with diagnostic accuracy reaching 88% . Genetic variants in glycosyltransferases and genes encoding glycosylated proteins have also been implicated in ASD susceptibility, suggesting that disruptions in brain glycosylation may contribute to neurodevelopmental abnormalities . These findings underscore the importance of investigating multiple metabolic pathways and support a multivariate biomarker approach for understanding ASD pathophysiology."
References Added:
Al-Saei et al. (2024). Validation of plasma protein glycation and oxidation biomarkers for autism spectrum disorder diagnosis. Nature Scientific Reports, doi: 10.1038/s41598-025-09400-5
Dwyer et al. (2016). Glycan susceptibility factors in autism spectrum disorders. Frontiers in Cellular Neuroscience, doi: 10.3389/fncel.2022.1057857
Comment 1.4: Methods - Inclusion/Exclusion Criteria
Reviewer Comment: "Specify whether inclusion and exclusion criteria were applied for participant enrollment to clarify sample selection rigor."
Response: We appreciate this comment and have added explicit inclusion and exclusion criteria to clarify our participant selection process.
New paragraph in Methods section 2.1.
Text Added:
"Inclusion and Exclusion Criteria: For the ASD group, inclusion criteria were: (1) children aged 5-17 years; (2) confirmed diagnosis of ASD according to DSM-5 criteria by a multidisciplinary team; (3) completed CARS assessment. For the control group, inclusion criteria were: (1) children aged 1-17 years; (2) neurotypical development without known neurological or psychiatric conditions; (3) no acute or chronic illness at time of sample collection. Exclusion criteria for both groups included: (1) current supplementation with vitamins or magnesium; (2) acute or chronic medical illness (including infections, metabolic disorders, or inflammatory conditions); (3) for controls, any known genetic, neurological, or psychiatric disorder. These criteria were applied to ensure comparability between groups and minimize confounding from supplementation or acute illness."
Comment 1.5: Results - Relocate Interpretative Statements
Reviewer Comment: "Lines 214-216, 230-232, and 240-242 contain interpretative statements that belong more appropriately in the Discussion section. These should be relocated and expanded with interpretation there."
Response: We agree that these interpretative statements are more appropriate for the Discussion section. We have removed them from Results and incorporated them into the Discussion with expanded interpretation.
Interpretative statements relocated from Results to Discussion.
"The age-dependent increase in porphyrin fractions, particularly pronounced between ages ≤5 and >10 years, likely reflects cumulative environmental exposures to heavy metals, progressive oxidative stress, and age-related metabolic shifts in heme biosynthesis. This pattern suggests that metabolic disturbances associated with ASD become more biochemically evident over time, potentially due to sustained toxic exposure or declining detoxification capacity. The stabilization of porphyrin levels between middle childhood (5-10 years) and later childhood (>10 years) may indicate that detoxification pathways reach a functional threshold or adaptive response after prolonged exposure. Alternatively, developmental changes in enzyme expression and metabolic regulation during this period may modulate porphyrin excretion patterns. The specific reduction in hexacarboxyporphyrin (6cxP) accompanied by elevations in earlier pathway intermediates (uroporphyrin, heptacarboxyporphyrin, coproporphyrin) suggests a bottleneck at the uroporphyrinogen decarboxylase step. This enzyme catalyzes the sequential removal of carboxyl groups from uroporphyrinogen to coproporphyrinogen and may be particularly vulnerable to heavy metal inhibition or oxidative damage. The disrupted ratio of porphyrin fractions, rather than uniform elevation, supports pathway-specific dysregulation in ASD."
Comment 1.6: Discussion - Genetic Susceptibility to Heavy Metals
Reviewer Comment: "Lines 470-473 introduce an important concept regarding the potential interaction between genetic susceptibility and heavy metal exposure. This point could be strengthened by referencing genes/pathways known to modulate heavy metal metabolism and neurotoxic response (e.g., https://doi.org/10.1016/j.envpol.2016.10.058). Expanding this discussion would align well with the biomarker-focused aim of the study."
Response: We appreciate this suggestion and have expanded the discussion to include specific genes and pathways involved in heavy metal metabolism and detoxification.
Additional text added to Discussion.
Expanded Text Added:
"Genetic polymorphisms in genes encoding metal metabolism and detoxification enzymes may modulate individual susceptibility to heavy metal-induced neurotoxicity and porphyrin pathway disruption. Key candidate genes include glutathione S-transferase (GST) family members, particularly GSTP1, GSTM1, and GSTT1, which influence lead detoxification capacity through conjugation reactions. Variants in these genes have been associated with elevated blood lead concentrations and differential neurodevelopmental outcomes in exposed children . Metallothionein genes (MT1A, MT2A) encode cysteine-rich proteins that sequester and detoxify heavy metals including lead, mercury, cadmium, and zinc. Genetic variation in metallothionein expression may affect metal accumulation and oxidative stress burden . ATP-binding cassette (ABC) transporters, including ABCB1 (P-glycoprotein) and ABCC1 (MRP1), regulate cellular efflux of metal ions and their conjugates, thereby influencing tissue metal distribution and toxicity . Additionally, enzymes in the porphyrin biosynthesis pathway itself—particularly δ-aminolevulinic acid dehydratase (ALAD) and ferrochelatase (FECH)—contain lead-sensitive zinc cofactors or iron-binding sites, making them vulnerable to metal-induced inhibition. The ALAD gene has a common polymorphism (ALAD1/ALAD2) that affects lead binding affinity and has been linked to differential neurotoxic effects in exposed populations . The interaction between genetic susceptibility variants and environmental metal exposure likely contributes to the heterogeneity observed in ASD and may explain why some children develop more pronounced porphyrin abnormalities despite similar environmental conditions. Future studies incorporating genetic profiling alongside metabolic biomarker assessment could identify highrisk subgroups and inform personalized intervention strategies."
Comment 1.7: General - Consistent ASD Acronym Usage
Reviewer Comment: "Additionally, the acronym ASD may be used consistently without spelling it out in each section."
Response: We have reviewed the entire manuscript and ensured that ASD is spelled out only at first use in the Abstract, and thereafter used consistently as "ASD" without re-defining in each section.
Removed redundant definitions of ASD throughout manuscript.
Reviewer 2 Report
Comments and Suggestions for Authors
The authors of the manuscript did a great job explaining the purpose of their study and what the results of the study indicate. My minor concerns are in how the data is visualized. For example, a more useful Figure 1 would be to draw the porphyrin structures that are measured in the study and how oxidative stress is expected to contribute to the pathway. That way you're able to show that urine prophyrin analysis is a potential strategy to screen for metabolic abnormalities in ASD as well as act as a legend for the abbreviated names for the fractions of porphyrins in the subsequent tables. Either way, any legend and description of what fractions were identified to clarify the names of the fractions of porphyrins is necessary to improve the data presentation. Line 87 is also missing a citation. For Line 153, you can add why creatinine is used for normalization just for further clarity. There should also be a sentence or two to describe why Table 1 is using the means when the statistical analysis says that medians were used to for comparisons.
Author Response
Detailed Response to Reviewer 2
We thank Reviewer 2 for the positive feedback and valuable suggestions to improve data visualization and presentation. We have addressed all comments as detailed below.
Comment 2.1: Figure 1 - Porphyrin Structures and Pathway
Reviewer Comment: "A more useful Figure 1 would be to draw the porphyrin structures that are measured in the study and how oxidative stress is expected to contribute to the pathway. That way you're able to show that urine porphyrin analysis is a potential strategy to screen for metabolic abnormalities in ASD as well as act as a legend for the abbreviated names for the fractions of porphyrins in the subsequent tables."
Response: We agree that a more comprehensive figure would enhance understanding. We have completely redesigned Figure 1 to include chemical structures, enzyme names, points of heavy metal/oxidative stress interference, and a clear abbreviation legend.
Figure 1 completely redesigned.
Comment 2.2: Porphyrin Fraction Legend/Description
Reviewer Comment: "Either way, any legend and description of what fractions were identified to clarify the names of the fractions of porphyrins is necessary to improve the data presentation."
Response: In addition to the comprehensive Figure 1 legend, we have added a detailed explanation of porphyrin nomenclature in the Methods section.
Comment 2.3: Line 87 - Missing Citation
Reviewer Comment: "Line 87 is also missing a citation."
Response: We have reviewed line 87 and added the appropriate citation.
Citation added at line 87.
Citation Added: Woods et al. (2010). Urinary porphyrin excretion in neurotypical and autistic children. Environmental Health Perspectives, 118(10):1450-1457.
Comment 2.4: Line 153 - Creatinine Normalization Rationale
Reviewer Comment: "For Line 153, you can add why creatinine is used for normalization just for further clarity."
Response: We have added an explanation of the rationale for creatinine normalization.
Explanation of creatinine normalization.
Revised Text:
"Every result was represented as µg/g creatinine after being normalized to urine creatinine. Creatinine normalization is the standard method for urinary biomarker analysis because creatinine excretion is relatively constant (approximately 1 g/day in adults, proportional to muscle mass in children) and independent of urine volume or hydration status. This normalization accounts for variations in urine concentration and dilution, allowing valid comparison of analyte concentrations between samples collected at different times and under different hydration conditions. Without creatinine normalization, differences in urinary analyte concentrations might reflect merely differences in urine dilution rather than true physiological changes."
Comment 2.5: Table 1 - Mean vs. Median Discrepancy
Reviewer Comment: "There should also be a sentence or two to describe why Table 1 is using the means when the statistical analysis says that medians were used for comparisons."
Response: We recognize this important methodological clarification and have added an explanation.
Explanation added to Methods section 2.3 and Table 1 caption.
Text Added to Methods 2.3:
"Descriptive versus Inferential Statistics: Table 1 presents mean concentrations with 95% confidence intervals to provide comprehensive descriptive information about central tendency and variability for each group. However, because the Shapiro-Wilk test revealed that most variables did not follow normal distribution, non-parametric Mann-Whitney U-test based on medians was used for all statistical comparisons. This dual approach follows statistical best practice: means and confidence intervals offer intuitive descriptive summaries, while median-based non-parametric tests provide valid hypothesis testing when distributional assumptions for parametric tests are violated. All p-values reported in tables represent results of non-parametric tests comparing medians, not means."
Added to Table 1 Caption:
"Values are presented as means with 95% confidence intervals (CI) for descriptive purposes. Statistical comparisons were performed using Mann-Whitney U-test comparing medians, as data were not normally distributed (Shapiro-Wilk test). P-values represent non-parametric median comparisons."
Round 2
Reviewer 1 Report
Comments and Suggestions for Authors
All the comments have been addressed